# Genome-Wide Comparative Analysis of *Lactiplantibacillus pentosus* Isolates Autochthonous to Cucumber Fermentation Reveals Subclades of Divergent Ancestry

**DOI:** 10.3390/foods12132455

**Published:** 2023-06-23

**Authors:** Clinton A. Page, Ilenys M. Pérez-Díaz, Meichen Pan, Rodolphe Barrangou

**Affiliations:** 1United States Department of Agriculture, Agricultural Research Service, SEA Food Science and Market Quality and Handling Research Unit, 322 Schaub Hall, Box 7624, Raleigh, NC 27695-7624, USA; 2Department of Food, Bioprocessing and Nutrition Sciences, North Carolina State University, 322 Schaub Hall, Box 7624, Raleigh, NC 27695-7624, USA

**Keywords:** *Lactiplantibacillus pentosus*, comparative genomics, *Lactiplantibacillus plantarum*, cucumber fermentation, starter culture

## Abstract

*Lactiplantibacillus pentosus*, commonly isolated from commercial cucumber fermentation, is a promising candidate for starter culture formulation due to its ability to achieve complete sugar utilization to an end pH of 3.3. In this study, we conducted a comparative genomic analysis encompassing 24 *L. pentosus* and 3 *Lactiplantibacillus plantarum* isolates autochthonous to commercial cucumber fermentation and 47 lactobacillales reference genomes to determine species specificity and provide insights into niche adaptation. Results showed that metrics such as average nucleotide identity score, emulated Rep-PCR-(GTG)_5_, computed multi-locus sequence typing (MLST), and multiple open reading frame (ORF)-based phylogenetic trees can robustly and consistently distinguish the two closely related species. Phylogenetic trees based on the alignment of 587 common ORFs separated the *L. pentosus* autochthonous cucumber isolates from olive fermentation isolates into clade A and B, respectively. The *L. pentosus* autochthonous clade partitions into subclades A.I, A.II, and A.III, suggesting substantial intraspecies diversity in the cucumber fermentation habitat. The hypervariable sequences within CRISPR arrays revealed recent evolutionary history, which aligns with the *L. pentosus* subclades identified in the phylogenetic trees constructed. While *L. plantarum* autochthonous to cucumber fermentation only encode for Type II-A CRISPR arrays, autochthonous *L. pentosus* clade B codes for Type I-E and *L. pentosus* clade A hosts both types of arrays. *L. pentosus* 7.8.2, for which phylogeny could not be defined using the varied methods employed, was found to uniquely encode for four distinct Type I-E CRISPR arrays and a Type II-A array. Prophage sequences in varied isolates evidence the presence of adaptive immunity in the candidate starter cultures isolated from vegetable fermentation as observed in dairy counterparts. This study provides insight into the genomic features of industrial *Lactiplantibacillus* species, the level of species differentiation in a vegetable fermentation habitat, and diversity profile of relevance in the selection of functional starter cultures.

## 1. Introduction

Diverse lactobacilli and *Leuconostoc* species are frequently isolated from vegetable fermentations, including cabbage, cucumber, green tomato, green bean, okra, olive, and onion, making them relevant for the formulation of starter cultures [1,2,3,4,5,6,7,8,9,10,11]. Starter cultures encompassing *Lactiplantibacillus plantarum* and *Lactiplantibacillus pentosus* are often used in the fermentation of blanched garlic, carrot, cucumber, olives kimchi, and kale, as well as cereal grains [5,12,13,14,15,16,17,18]. Fermentative bacteria such as lactobacilli are implicated in the conversion of sugars to lactic acid and the production of ascorbic acid, glutathione, flavonoid aglycones, and distinguishable volatile compounds, as well as in enhancing total antioxidant activity [7,19]. Fermentative bacteria also reduce total amino acid concentration in fermented vegetables, leading to the production of esters, alcohols, and aldehydes presumably involved in flavor development [20]. The metabolic activity of fermentative bacteria confers complex chemical and biochemical characteristics to fermented vegetables of relevance for preservation and organoleptic attributes.

The growth of *L. plantarum* and *L. pentosus* in cucumber fermentation favors the preservation of pickle quality. Such lactobacilli convert sugars to lactic acid exclusively in cucumber fermentation, which favors the prevention of bloater defects from the accumulation of carbon dioxide inside the fruits [21]. *L. plantarum* and *L. pentosus* lead a rapid reduction in pH relative to competing microbes and tolerate pH as low as 3.1 [22]. Additionally, it is speculated that starter cultures autochthonous to vegetable fermentations could arrest the growth of spoilage-associated microbes through competitively excluding energy sources or producing bacteriocins [23]. This is relevant for the development of low-salt cucumber fermentation, in which spoilage-associated microbes can accumulate after the completion of sugar conversion, elevate pH, and jeopardize the microbial stability of the fermented product [24,25,26]. Thus, the informed design of lactobacilli starter culture for cucumber fermentation can function in enhancing the performance of fermentation as well as preventing spoilage.

*L. pentosus* and *L. plantarum* are closely related lactic acid bacteria (LAB) originally defined by the ability of the former to metabolize pentose sugars into lactic acid [27]. The species definition was later refined to specify metabolism of the pentose sugar D-xylose as well as glycerol as characteristic of *L. pentosus* [28]. Subsequent research confounded the usefulness of D-xylose and glycerol utilization as a phenotypic test, as strains belonging to both species were found capable of such metabolic activity [29]. While DNA–DNA hybridization can discriminate between *L. plantarum* and *L. pentosus* strains [29,30], the 16S rDNA sequence-based phylogeny construction is often insufficient to discriminate the species [31]. The construction of a phylogenetic tree based on a single housekeeping gene sequence, including *recA*, *dnaK,* and *pheS*, can discriminate between *L. plantarum* and *L. pentosus* but is limited in assessing intraspecies genetic diversity [32]. Random Amplified Polymorphic DNA (RAPD)-PCR combined with analysis of banding patterns, was sufficient to correctly group seven isolates (1.2.13, 3.8.24, 1.2.11, 1.8.6, 1.8.9, 3.2.37, and LA0445) with *L. pentosus* reference strains while discriminating among themselves [32]. The same analysis also grouped *L. plantarum* isolates with *L. plantarum* reference strains [32]. However, the availability of whole genome sequences enables comparative genomic analysis which is useful in discriminating among the two species and in identifying putative and divergent functions. Calculation of average nucleotide identity (ANI) for genome sequences against references can quickly identify the affiliation of the harboring organism to a given species [33,34]. The identification of divergent or unique putative functions informs the selection of strains to serve as starter cultures that can exert a function within the intrinsic microbiome.

This study analyzed genome sequences from *L. plantarum* and *L. pentosus* isolates autochthonous to commercial cucumber fermentations that were selected based on multiple criteria. Most of the bacteria were selected for whole genome sequence comparative analysis based on their ability to rapidly reduce pH in multiple cucumber juice treatments varying in starting pH, temperature, and salinity [32]. *L. plantarum* T1R2b and 3.2.8 were selected for their ability to produce an unusually slimy, exopolysaccharide (EPS)-like matrix in cucumber fermentation [35]. The primary goal of the study was to determine species-specific genomic content and assess diversity between and within species. The approach was to apply comparative genome sequence analysis focused on the metrics for structural genome features and phylogeny, determination of ANI scores, computed MLST, emulated Rep-PCR, prophage coding, and CRISPR arrays to define diversity as relevant for cucumber fermentation.

## 2. Materials and Methods

*L. plantarum* and *L. pentosus* isolates and genome sequences used in this study: Twenty-seven pure cultures were included in this study, of which twenty-two were isolated from a commercial cucumber fermentation tank filled with recycled cover brine in 2009 as described by Pérez-Díaz et al. [32]. The twenty-two cultures include *L. plantarum* 3.2.8 and 7.8.4, and twenty *L. pentosus* cultures. *L. pentosus* 7.8.2 and 7.2.20 were isolated from an independent commercial cucumber fermentation tank filled with fresh cover brine in 2010 [32]. *L. pentosus* LA0445 is of cucumber fermentation origin isolated in 1983 and MU045 is a mutant derivative of this strain [36,37]. *L. plantarum* T1R2b was isolated in 2020 from a low-salt cucumber fermentation that produced an irregular slimy cover brine [38].

The sequencing, assembly, and annotation, along with growth conditions, of the genomic sequences corresponding to the *L. plantarum* and *L. pentosus* cultures autochthonous to cucumber fermentation and included in this study are described by Page & Pérez-Díaz [39] and available from the National Center for Biotechnology Information (NCBI) database under BioProject Accession Number PRJNA674638. GenBank accession numbers corresponding to the genome sequences derived from the autochthonous and allochthonous lactic acid bacteria included in this study are listed in Table 1.

Categorical description of genome sequences: The descriptive data of the genome sequence features and coding gene type shown in Figure 1 were determined in the IMG system [40] (analysis was run in July 2022) using the genome browser function and modifying the output table configuration for the data shown. Significant difference between the *L. pentosus* and *L. plantarum* genome sequences was determined using a T-test performed in Microsoft Excel (2019) for each feature or gene coding type, individually.

ANI score calculation for the *L. plantarum* and *L. pentosus* genome sequences: Calculations of the ANI scores between an *L. pentosus* and an *L. plantarum* were carried out in the IMG system [40] (analysis was run in July 2022). Estimated ANI scores for pairwise comparisons of *L. pentosus* or *L. plantarum* homolog genomes were calculated in Kbase via the FastANI v0.13 automated pipeline [41] (analysis run in March 2022).

Construction of phylogenetic trees: Three distinct phylogenetic trees were constructed for definitive culture identification and comparison of species boundaries. A phylogenetic tree was constructed based on a computer emulated Rep-PCR using the (GTG)_5_ primer [42] for the genome sequences of several lactic acid bacteria whose genome sequences are described in Table 1. The metadata for the genome sequences described in Table 1 that are autochthonous to cucumber may be found in Page & Perez-Diaz [39]. Amplicons from a computed Rep-PCR-(GTG)_5_ were generated with Geneious software (v. 2021.1.1, www.geneious.com, analysed in June 2021, accessed on 1 June 2023) for each genome sequence of interest. The band patterns produced for each genome sequence were compared using Bionumerics software v. 7.6.3 (www.applied-maths.com, accessed on 1 June 2023). Similarity matrices of the densitometric curves from each sample were calculated using Pearson correlation coefficients and clustered via unweighted pair group method with arithmetic averages (UPGMA).

A second phylogenetic tree was produced using a computer-emulated MLST run in AutoMLST [43] (analysis run in December 2021) based on the 79 core genes described in Appendix A, which were automatically selected by the software. This analysis was focused on the clustering of autochthonous and allochthonous *L. plantarum* and *L. pentosus* genomes, whose accession numbers are listed in Table 1. The genome sequence of *Pediococcus damnosus* DSM20331 was selected by AutoMLST as the tree root.

A phylogenetic tree was generated in PATRIC (analysis run in December 2021) [44] via the Codon Tree method, which employs RaxML to generate phylogenetic distances between sequences of *L. pentosus* genomes exclusively [45]. Accession numbers for the *L. pentosus* genomes used, from cucumber fermentation autochthonous and allochthonous isolates, are listed in Table 1. No duplications or deletions were allowed for the analysis. The 587 ORFs listed in Appendix A were used for the construction of the phylogenetic tree.

Tree diagrams were generated for the computed MLST and multiple ORFs phylogenetic analysis outputs with Interactive Tree of Life (iTOL) v.6.5.7 [46] in June 2022. Default parameters were used for the construction of trees.

CRISPR-Cas system identification and characterization: The CRISPRCasTyper pipeline was used to scrutinize the genome sequences of interest [47]. CRISPR spacers were extracted and visualized using the CRISPRViz pipeline [48]. The extracted spacer sequences were passed to BLASTn searches in the NCBI nucleotide database to identify potential protospacer sequences. The positive BLAST hits had an e-value smaller than 1 × 10^−3^ and an identity score greater than 85%. The flanking regions of the protospacer matches were extracted and aligned to identify putative protospacer adjacent motifs (PAM) (Appendix A). The BLAST-webserver search was conducted against the nucleotide collection [49] in December 2021.

Identification of putative prophage sequences: A survey of putative prophage sequences was conducted among the genome sequences derived from the cucumber-autochthonous *L. plantarum* and *L. pentosus* using the Phage Search Tool enhanced release (PHASTER, analysis run in February 2022) [50,51]. The multiple contigs option was selected for the prophage sequence analysis and default parameters were applied for the other options in the service. Accession numbers for the lactobacilli genome sequences used are listed in Table 1.

## 3. Results and Discussion

Features of the *L. pentosus* and *L. plantarum* genome sequences: The genome sizes of most LAB are between 1.8 and 3.4 megabases [52], which ranges on the lower end of the bacterial genome spectrum spanning approximately 500 kilobases to 12 megabases [53]. LAB are thought to be highly specialized given their evolutionary path and historical association with select niches over the course of human association and use for fermentation processes in relatively nutrient-rich habitats [52]. Such bacteria are known for their ability to use horizontal gene transfer for adaptation [54]. Generally, bacteria exhibit mutational bias that deletes superfluous sequences, which produces smaller genomes with a greater degree of specialization [55].

The *L. pentosus* genomes studied here ranged in size from 3.59 to 3.83 Mbp with 46% GC content, while the *L. plantarum* genome sequences were found to be significantly smaller at 3.38 to 3.48 Mbp with 44% GC content (Figure 1). The range of such genome sizes and GC percentages are consistent with the ranges reported in the PATRIC database for each species [19]. The benefits derived by *L. pentosus* from the additional genes relative to *L. plantarum* and the metabolic versatility associated with such genes could confer a competitive advantage [56,57]. Figure 1 shows that the predicted signal peptides and proteins, specifically transmembrane proteins, are more abundant in the *L. pentosus* genome as compared to those in the *L. plantarum* counterpart. It is documented that *L. pentosus* is more commonly isolated from cucumber and olive fermentations than *L. plantarum* [2,10]. Such observations suggest that the larger *L. pentosus* genome may be advantageous in sensing its habitat, fermentation in this case.

ANI is widely used to determine whether multiple genomes belong to the same species [34,58]. As expected, the calculation of ANI values from a comparison of a *L. pentosus* and *L. plantarum* genome sequences was approximately 80%, significantly less than the 95% cut-off score for species membership (Figure 2A). The ANI calculations described here were generated through pairwise comparisons between an autochthonous *L. plantarum* or *L. pentosus* genome sequence and a homolog genome sequence or a reference genome (Figure 2A). Contrary to *L. plantarum* autochthonous to cucumber fermentation (Figure 2B), the calculation of varied ANI values above the 95% threshold among the *L. pentosus* counterparts suggested a degree of intraspecies variation worth studying for discerning adaptive diversity of value in the optimization of starter cultures for cucumber fermentations (Figure 2C).

The 27 isolates included in this study were previously identified through the alignment of the 16S rDNA partial sequence to those of reference strains and the *recA* amplicon size as described by Torriani et al. [10,31]. The identification of 26 out of the 27 isolates in the study was confirmed via the ANI scores calculated as well as via the whole genome sequence alignments to reference genomes in the NCBI and the IMG/M databases, the exception being *L. pentosus* 7.2.4, which was initially classified as *L. plantarum* [10].


***L. pentosus* and *L. plantarum* intraspecies genetic diversity as defined through the construction of phylogenetic trees:**


The simulation of Rep-PCR-(GTG)_5_ through emulated banding patterns from whole genome sequences discriminated between the *L. pentosus* and *L. plantarum* autochthonous to cucumber fermentation (Figure 3) and replicated the separation previously observed by Pérez-Díaz et al. [32] for *L. pentosus* LA0445 from six commercial fermentation isolates of the same species (1.2.13, 3.8.24, 1.2.11, 1.8.6, 1.8.9, and 3.2.37) using benchtop Rep-PCR-(GTG)_5_. The *L. pentosus* strain separation identified using computer-emulated and benchtop Rep-PCR-(GTG)_5_ phylogenetic clustering was also detected in AutoMLST analysis including 79 common ORFs (Figure 4A) and using PATRIC-based clustering aligning 587 common ORFs (Figure 4B). The four analyses suggest at least two distinct *L. pentosus* clades, A and B, which include identical clusters of isolates. *L. pentosus* clade A includes isolates collected in North Carolina from fermentation day one to day thirty, while *L. pentosus* clade B includes three isolates collected from fermentation day seven or later. Among these are LA0445, which was derived from a commercial fermentation conducted in 1983, as well as MU045, a derivative of the former deficient in malic acid decarboxylation [59]. Two *L. pentosus* isolated from cucumber fermentation conducted in North Carolina (2009) also belong to clade B, 7.2.4 and 7.2.11. Two *L. pentosus* isolated from a cucumber fermentation conducted in Minnesota (7.2.20 and 7.8.2), as well as multiple isolates collected on days seven and 14 of the Carolinian counterparts (7.2.15, 7.8.11, 7.8.46, 14.2.3, 14.2.16, and 14.8.42) did not sort consistently into clade A or B (Figure 4A).

A subsequent phylogenetic construction including only *L. pentosus* strains employed 1000 core ORFs and expanded the *L. pentosus* clade A described in Figure 4 to include isolates that did not sort into clade A or B in the previous trees (Figure 5). Clade A can be further subdivided into three subclades, A.I, A.II, and A.III. Subclade A.I includes twelve isolates, ten of which were collected on day one or three of fermentation. Subclade A.II is a group of two isolates, one each from the cucumber fermentations conducted in North Carolina and Minnesota, both isolated on day seven. Subclade A.III includes five *L. pentosus* isolated on day seven or 14 of the cucumber fermentation conducted in North Carolina. *L. pentosus* 7.8.2 was isolated from the cucumber fermentation performed in Minnesota and did not sort with isolates from clade A or B but is consistently associated with *L. pentosus* reference genomes from multiple sources (Figure 5). This phylogenetic analysis also included multiple strains derived from olive fermentation, which, with the exception of a single isolate associated with subclade A.I, clustered in several distinct clades not associated with *L. pentosus* clades A or B.

The three autochthonous *L. plantarum* isolates were associated with counterparts from multiple habitats based in the three phylogenetic analyses employing emulated Rep-PCR-(GTG)_5_, computed MLST data (AutoMLST) involving 79 common ORFs, and 587 common ORFs via the PATRIC Codon Tree builder (Figure 4). *L. plantarum* 3.2.8 and 7.8.4 share a common ancestor with the *L. plantarum* strain ZJ316, which is a human fecal isolate with reported probiotic properties [60]. *L. plantarum* T1R2b clustered into a separate branch from 3.2.8 and 7.8.4, which has common ancestry with *L. plantarum* B21, a fermented sausage isolate able to produce a broad spectrum bacteriocin B21AG, which is also a circular plantacyclin [61]. The *L. plantarum* T1R2b draft genomic DNA sequence, however, lacks the putative genes encoding for such bacteriocin [62]. It is relevant to note that two of the three *L. plantarum* autochthonous to cucumber fermentation are capable of prevailing in the native habitat and consistently produce slimy brines. Thus, exploitation of such *L. plantarum* isolates in cucumber fermentation is limited.

Occurrence and diversity of CRISPR-Cas immune systems: The hypervariable nature of CRISPR loci has been used in the past for the genotyping of various genera and species, including starter cultures and food pathogens, notably, *Streptococcus thermophilus* for the former [63,64] and *E. coli* and *Salmonella* for the later [65,66]. Here, we sought to exploit the hypervariable CRISPR arrays in the *L. pentosus* and *L. plantarum* genomes to discern intraspecies diversity and ancestry. Both type I-E and type II-A CRISPR-Cas systems were identified in the *L. pentosus* genomes, with canonical *cas* operon structures and occurrence of variable spacers in CRISPR arrays (Table 2). Distinct type II-A arrays were detected in *L. plantarum* isolates, as well as in *L. pentosus* 7.8.2 and the *L. pentosus* subclades A.I, A.II, and A.III. The fact that no type II-A spacers were shared between the subclades confirms the significance of the isolates’ clustering observed in phylogenetic analyses (Figure 6). The identified CRISPR arrays in the *L. plantarum* isolates did not share similar spacer sequences despite the close association of these isolates in phylogenetic trees. No CRISPR-Cas system was detected in the *L. plantarum* isolate T1R2b.

*L. pentosus* isolates 7.8.46 and 7.2.20, which belong to subclade A.II, shared common type I-E CRISPR arrays with subclade A.III but not with subclade A.I, suggesting that A.II shares more recent ancestry with A.III than subclade A.I (Figure 7). *L. pentosus* 7.8.46 and 7.2.20 also shared a type II-A array that was not detected in other *L. pentosus* isolates, further supporting the separation of these isolates from subclades A.I and A.III. Type I-E CRISPR arrays detected in the *L. pentosus* isolate 7.8.2 were distinct from other *L. pentosus* CRISPR sequences, which reflected the separation of strain 7.8.2 from other *L. pentosus* isolates described in phylogenetic analysis (Figure 8).

Four type I-E spacers were found to match known phages previously isolated from the same fermentation tanks (Figure 9) [10,26]. Unexpectedly, three separate spacers in *L. pentosus* 7.2.11 matched to two different phages, ϕJL-1 and ϕSha, while a fourth spacer in *L. pentosus* 7.2.4 targeted ϕSha (Figure 9). All protospacers were flanked by putative protospacer adjacent motif (PAM) sequence 5′-AAA-3′ on the 5′ end, reflecting CRISPR-mediated immunity in *L. pentosus* against these phages during industrial fermentation. This is consistent with the role of CRISPR-Cas systems as providers of adaptive immunity as originally demonstrated in dairy starter cultures [67].

Putative prophage profiles detected in the *L. pentosus* and *L. plantarum* genome sequences: It is estimated that 10% of the bacterial community in commercial cucumber fermentations is susceptible to bacteriophage infection, of which a fifth is attributed to *L. pentosus* and *L. plantarum* [68]. Phages of the *Myoviridae* and *Siphoviridae* are commonly isolated from vegetable fermentations [68,69]. Five intact prophage sequences of the *Siphoviridae* sp. were detected in 7 of the 24 *L. pentosus* isolates (Table 3). *Siphoviridae* ctu0P1 was found in the genomic DNA sequences of six *L. pentosus* isolated on days one and three of commercial fermentations, within subclade A.I. The genome sequences of the *L. pentosus* 14.2.3 includes two different *Siphoviridae*, ctu0P1 and ctk5O4, as well as a third phage with homology to *Siphoviridae* ctk5O4. Phages with homology to *Siphoviridae* ctu0P1 were also detected in two *L. plantarum* isolates, 7.8.4 and T1R2b. Two additional phages were detected in the genome sequences of *L. plantarum* 7.8.4 and another was found in T1R2b, all of which have homology to *Siphoviridae.* Together, these observations suggest the presence of grounded *Siphoviridae* phages in *L. plantarum* and *L. pentosus* autochthonous to commercial cucumber fermentations, which may be conferring immunity and serving as drivers of evolution and biogenesis of genetic information [70].

## 4. Conclusions

Given the inability of classical phenotypic tests and 16S rDNA-based methods to differentiate between *L. plantarum* and *L. pentosus*, the verification of multiple genomic-based methods can assist researchers in properly identifying and discriminating such species. The phylogenetic distribution of *L. pentosus* isolates from commercial cucumber fermentation suggests the co-existence of multiple *L. pentosus* strains in this habitat. The presence of Type I-E CRISPRs arrays in *L. pentosus* isolates from clade B that are homologous to identified phages provides evidence of adaptive immunity in candidate starter cultures. Such findings underscore the need to decipher the genetic composition of strains to be used as commercial starter cultures and to design formulations that can exert specific functional features relevant for complex industrial fermentations.

**Short version of the title:** comparative genomics of *L. pentosus*

Mention of a trademark or proprietary product does not constitute a guarantee or warranty of the product by the US Department of Agriculture or North Carolina Agricultural Research Service, nor does it imply approval to the exclusion of other products that may be suitable. USDA is an equal opportunity provider and employer.

## Figures and Tables

**Figure 1 foods-12-02455-f001:**
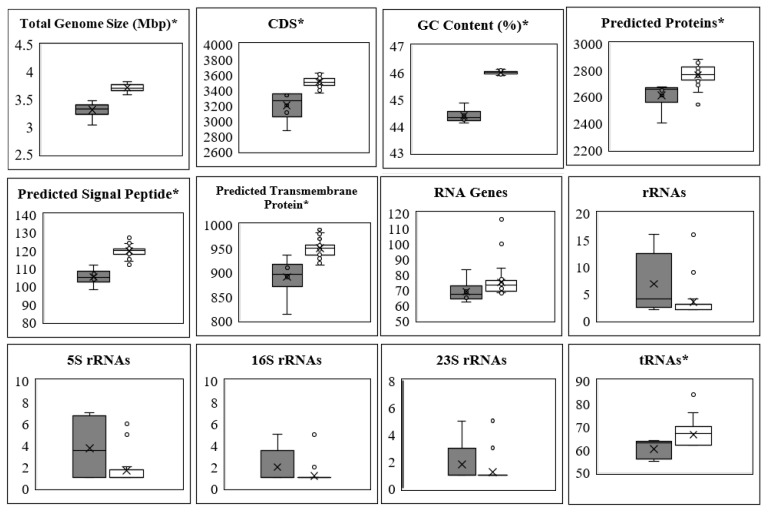
Descriptive data of the *Lactiplantibacillus plantarum* (■) and *Lactiplantibacillus pentosus* (□) genome sequence features and coding gene type. A significant difference (*p*-value < 0.05) was calculated for panels identified with an asterisk (*).

**Figure 2 foods-12-02455-f002:**
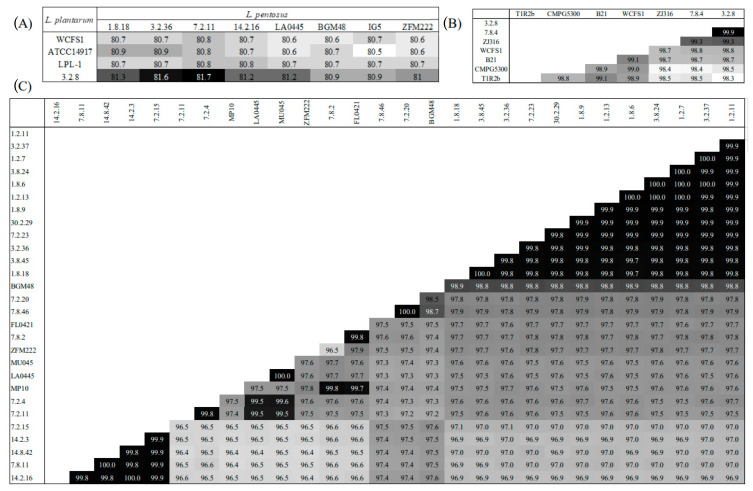
Heat maps displaying average nucleotide identity (ANI) for *L. pentosus* versus *L. plantarum* (**A**) and within *L. plantarum* (**B**) and *L. pentosus* (**C**) isolates versus a selection of reference genomic DNA sequences. Darker shading indicates elevated relative ANI scores between two isolates or reference strains.

**Figure 3 foods-12-02455-f003:**
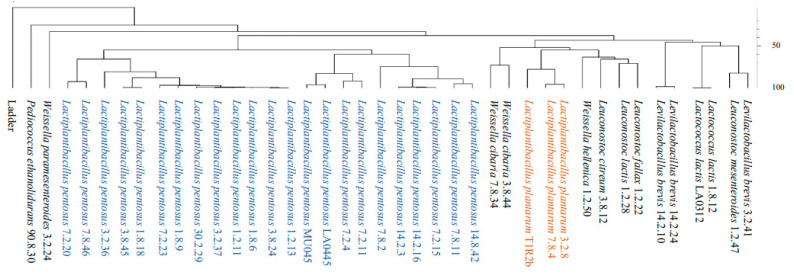
Computer-emulated Rep-PCR-(GTG)_5_ and banding pattern clustering analysis based on 40 genome sequences from selected *Lactobacillales. L. pentosus* and *L. plantarum* isolates are labeled blue and orange, respectively.

**Figure 4 foods-12-02455-f004:**
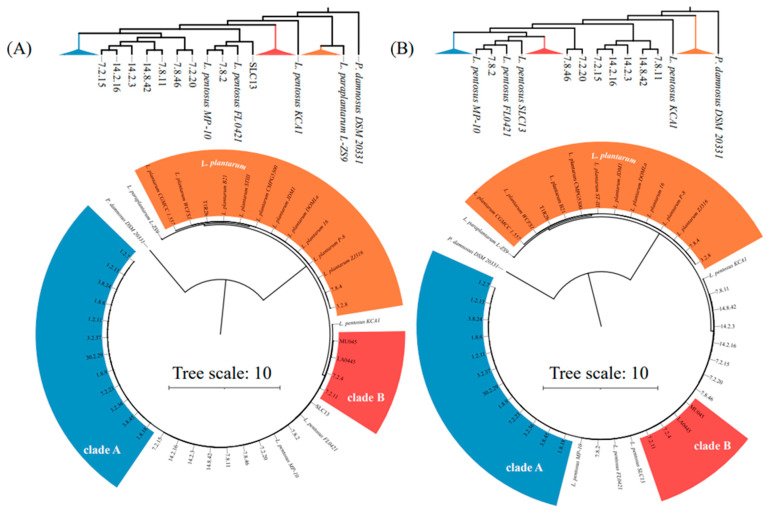
Phylogenetic trees, including all *L. plantarum* and *L. pentosus* isolates as well as multiple reference strains for each species based on MLST of sequences from 79 common ORFs (**A**) and the alignment of 587 common ORFs (**B**). Colored bands indicate *L. plantarum* isolates and reference strains and *L. pentosus* phylogenetic subgroups clade A and clade B. *Pediococcus damnosus* DSM 20331 was selected as the root for both trees based on AutoMLST automatic selection. Trees without scale are included in order to more clearly indicate branching, with matched colored triangles representing groupings in the circular trees.

**Figure 5 foods-12-02455-f005:**
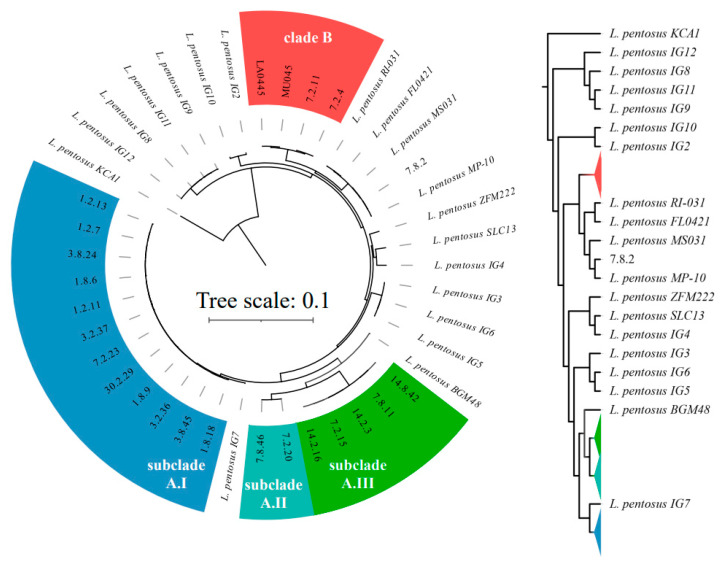
A phylogenetic tree based on 1000 core genes in *L. pentosus* autochthonous to cucumber and olive fermentations and reference strains. Subclades A.I, A.II, and A.III and clade B are indicated with superimposed color bands. *L. pentosus* KCA1 is the root for this tree. The rectangular tree diagram with collapsed nodes representing matching-colored clades is included to more clearly show individual branches.

**Figure 6 foods-12-02455-f006:**
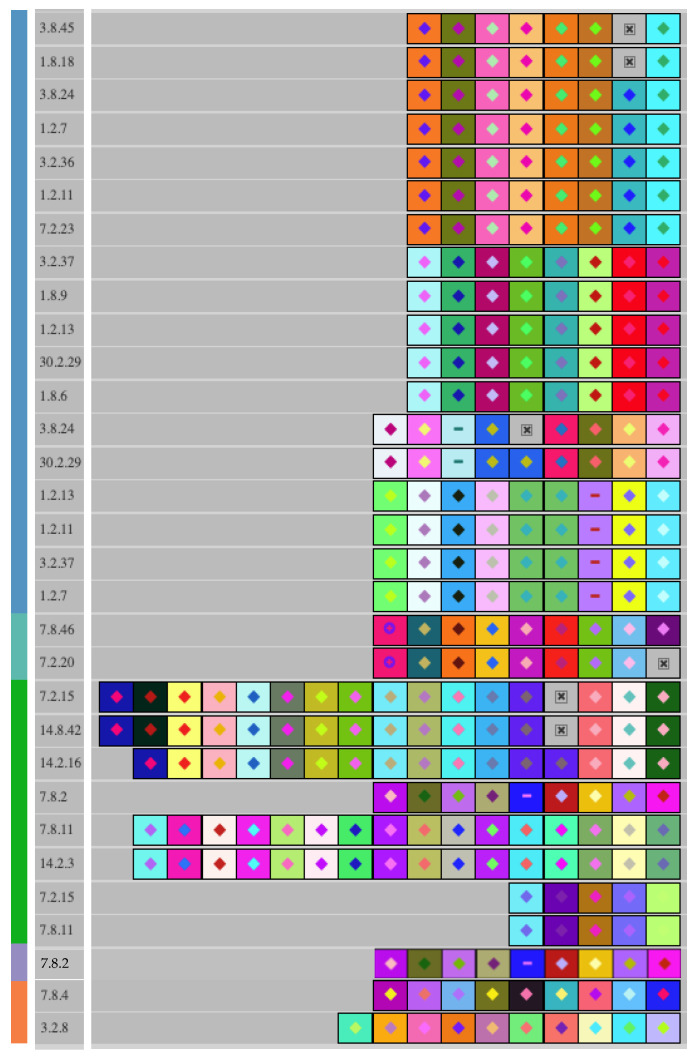
The Type II-A CRISPR spacers were extracted and aligned from *L. plantarum* (■) and *L. pentosus* autochthonous to commercial cucumber fermentation belonging to subclade A.I (■), subclade A.II (■), subclade A.II (■), or 7.8.2 (■). Differently colored boxes represent individual spacer sequences in each CRISPR array. Identically colored boxes in two or more isolates indicate a shared spacer sequence between those isolates.

**Figure 7 foods-12-02455-f007:**
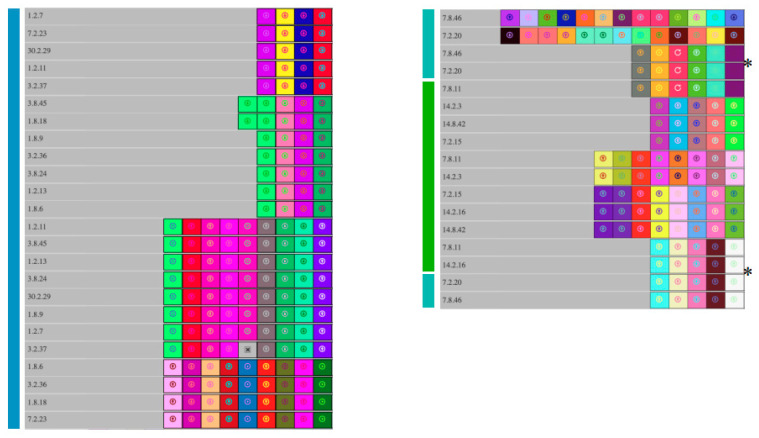
The Type I-E CRISPR spacers were extracted and aligned from the *L. pentosus* clade A genomes, subclade A.I (■), subclade A.II (■), and subclade A.III (■). All *L. pentosus* strains were isolated from commercial cucumber fermentations. Differently colored boxes represent individual spacer sequences in each CRISPR array. Identically colored boxes in two or more isolates indicate a shared spacer sequence between those isolates. Asterisks indicate CRISPR spacer arrays detected in separate subclades.

**Figure 8 foods-12-02455-f008:**
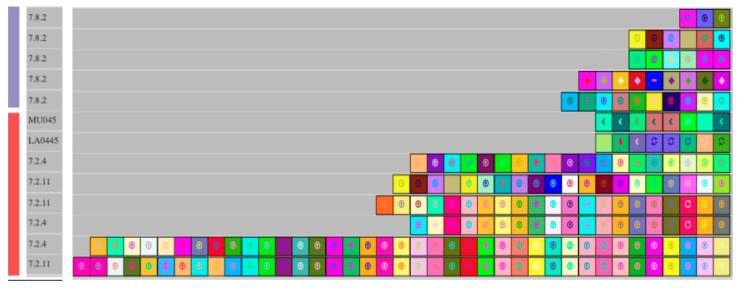
Type I-E CRISPR spacers extracted and aligned from *L. pentosus* clade B genomes (■) and the *L. pentosus* 7.8.2 genome (■). All *L. pentosus* strains were isolated from commercial cucumber fermentations. Differently colored boxes represent individual spacer sequences in each CRISPR array. Identically colored boxes in two or more isolates indicate a shared spacer sequence between those isolates.

**Figure 9 foods-12-02455-f009:**
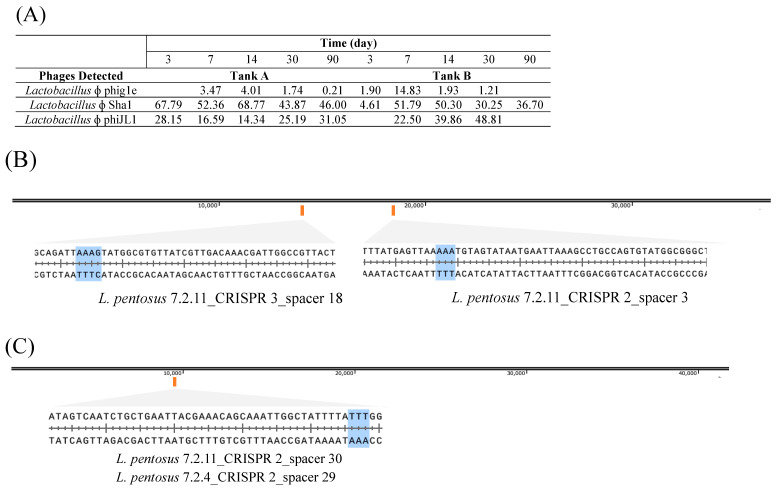
Incidence of phages (%) in industrial cucumber fermentations as determined via metagenomic analysis conducted by Pérez-Díaz et al. (2020) (**A**), and spacer matches to *Lactobacillus* ɸ phiJL1 (**B**) and *Lactobacillus* ɸ Sha1(**C**).

**Table 1 foods-12-02455-t001:** GenBank accession numbers for the genome sequences used in this study.

Strain	Accession Number	Strain	Accession Number
*Lactiplantibacillus plantarum*			
CMPG5300	GCA_000762955.1	JDM1	GCA_000023085.1
B21	GCA_000931425.2	DOMLa	GCA_000604105.1
WCFS1	GCA_000203855.	16	GCA_000412205.1
ZJ316	GCA_000338115.2	P-8	GCA_000392485.2
ATCC14917	GCA_000143745.1	7.8.4	GCA_018993285.1
LPL-1	GCA_002205775.2	3.2.8	GCA_018991195.1
ST-III	GCA_000148815.2	T1R2b	GCA_018993325.1
CGMCC1.557	GCA_001272315.2		
*Lactiplantibacillus pentosus*			
BGM48	GCA_002850015.1	1.8.6	GCA_018991355.1
FL0421	GCA_001188985.1	1.8.9	GCA_018993945.1
ZFM222	GCA_003627295.1	1.8.18	GCA_018993635.1
MP-10	GCA_900092635.1	3.2.37	GCA_018993495.1
KCA1	GCA_000271445.1	3.2.36	GCA_018993725.1
RI-031	GCA_002751855.1	3.8.24	GCA_018993345.1
MS031	GCA_016804305.1	3.8.45	GCA_018991565.1
SLC13	GCA_002211885.1	7.2.4	GCA_018991185.1
IG2	GCA_002993465.1	7.2.11	GCA_018991375.1
IG3	GCA_002993425.1	7.2.15	GCA_018993445.1
IG4	GCA_002993385.1	7.2.20	GCA_018993665.1
IG5	GCA_002993435.1	7.2.23	GCA_018993485.1
IG6	GCA_002993485.1	7.8.2	GCA_018991535.1
IG7	GCA_002993395.1	7.8.11	GCA_018993275.1
IG8	GCA_003702625.1	7.8.46	GCA_018991465.1
IG9	GCA_003702665.1	14.8.42	GCA_018993525.1
IG10	GCA_003702635.1	14.2.3	GCA_018991425.1
IG11	GCA_003702605.1	14.2.16	GCA_018993625.1
IG12	GCA_003702565.1	30.2.29	GCA_018991255.1
1.2.7	GCA_018991365.1	LA0445	GCA_018993965.1
1.2.11	GCA_018991285.1	MU045	GCA_018993585.1
1.2.13	GCA_018991345.1		
*Weissella* spp.		*Leuconostoc* spp.	
*W. paramesenteroides* 3.2.24	GCA_018991665.1	*Leu. citreum* 3.8.12	GCA_018991475.1
*W. cibaria* 7.8.34	GCA_025770435.1	*Leu. lactis* 1.2.50	GCA_018993865.1
*W. cibaria* 3.8.44	GCA_018993825.1	*Leu*. *fallax* 1.2.22	GCA_018993745.1
*W. sagaensis* 1.2.50	GCA_018993865.1	*Leu. mesenteroides* 1.2.47	GCA_018991785.1
*Pediococcus* spp.		*Lactococcus* spp.	
*P. ethanolidurans* 90.8.30	GCA_018993685.1	*Lc. lactis* LA0312	GCA_018991605.1
*P. damnosus* DSM 20331	GCA_001437255.1	*Lc. lactis* 1.8.12	GCA_018991645.1
*Levilactobacillus* spp.			
*Lev. brevis* 14.2.24	GCA_018993885.1	*Lactiplantibacillus paraplantarum* L-ZS9	GCA_001443645.1
*Lev. brevis* 14.2.10	GCA_018991505.1		
*Lev. brevis* 3.2.41	GCA_018991675.1		

**Table 2 foods-12-02455-t002:** Incidence of CRISPR loci in the genomic DNA sequences of *L. plantarum* and *L. pentosus* autochthonous to cucumber fermentation and reference strains by type.

	Genomic DNA Sequence Identification							
	*L. plantarum*			*L. pentosus*				
	3.2.8	7.8.4	T1R2b	Subclade A.I	Subclade A.II	Subclade A.III	Clade B	7.8.2
CRISPR loci	1	1	0	4 ± 1	4	4 ± 1	2 ± 1	5
Type I-E	0	0	0	2	3	2	2 ± 1	4
Type II-A	1	1	0	2 ± 1	1	1 ± 1	0	1

**Table 3 foods-12-02455-t003:** Closest matches to the putative prophages found in the *L. pentosus* and *L. plantarum* genomic sequences in the NCBI database.

Closest Homolog Identity (>95%)	*L. plantarum* Isolates Autochthonous to Cucumber Fermentation																							
	3.2.8													7.8.4						T1R2b				
*Siphoviridae* sp. ctu0P1														X										
*Siphoviridae* sp. ctyPQ2														X										
*Siphoviridae* sp. ct97I3														X										
*Siphoviridae* sp. ctk5O4																								
*Siphoviridae* sp. ctUZT2																				X				
Closest Homolog Identity (>95%)	***L. pentosus* Isolates Autochthonous to Cucumber Fermentation**																							
	**Subclade A.I**												**Subclade A.II**		**Subclade A.III**					**Clade B**				
	1.2.7	1.2.13	3.8.24	1.8.6	1.2.11	3.2.37	30.2.29	1.8.9	7.2.23	3.2.36	3.8.45	1.8.18	7.8.46	7.2.20	14.2.3	7.2.15	14.2.16	14.8.42	7.8.11	7.2.11	7.2.4	LA0445	MU0045	7.8.2
*Siphoviridae* ctu0P1															X	X	X	X	X		X			
*Siphoviridae* ctyPQ2																								
*Siphoviridae* ct97I3																								
*Siphoviridae* ctk5O4															X									
*Siphoviridae* ctUZT2																								

## Data Availability

The data used to support the findings of this study can be made available by the corresponding author upon request.

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
