# Peer review of "Genome-Wide Comparative Analysis of Lactiplantibacillus pentosus Isolates Autochthonous to Cucumber Fermentation Reveals Subclades of Divergent Ancestry"

_foods, 2023, doi:10.3390/foods12132455_

Round 1

Reviewer 1 Report

Dear Editors and authors, 

1-The abbreviation of the scientific name of living organisms is very important during the writing of manuscripts and studies. After updating the naming of lactic acid bacteria, the appropriate abbreviation must be chosen for each genus, and the letter L can be used for all genera. Therefore, the  : Lactiplantibacillus genus abbreviation should be Lcp.

2- Introduction to the manuscript needs to add some modern scientific references about the use of lactic acid bacteria in fermented vegetables, such as

Al-Sahlany, S. T., & Niamah, A. K. (2022). Bacterial viability, antioxidant stability, antimutagenicity and sensory properties of onion types fermentation by using probiotic starter during storage. Nutrition & Food Science, 52(6), 901-916.

Ashaolu, T. J., & Reale, A. (2020). A holistic review on Euro-Asian lactic acid bacteria fermented cereals and vegetables. Microorganisms, 8(8), 1176.

Lorn, D., Ho, P. H., Tan, R., Licandro, H., & Waché, Y. (2021). Screening of lactic acid bacteria for their potential use as aromatic starters in fermented vegetables. International Journal of Food Microbiology, 350, 109242.

3-The number of bacterial strains does not match in the explanation, the authors say we used 27 strains, but when detailing it, it is only 24 strains, while Table 1 contains 37 strains.

4-Many of the strains mentioned in Table 1 were not mentioned by the authors, what are their sources, and where did they come from in the study? such as Weissella spp. , Leuconostoc spp. , and.............., etc. 

5-The authors did not mention how these bacteria grew ? What culture medium was used for development? What were the growth conditions?

6- There is no mention in the work methods of the statistical analysis of the results in the results chapter.

7-Figure No. 6 contains many colors whose meaning was not mentioned or explained. Only 5 colors were mentioned.

8-The same previous note applies to Figures 7 and 8.

9-Conclusions should be rewritten again because they contain many results. Results should be deleted.

The language of the manuscript is good. 

Reviewer 2 Report

General/major comments

In this paper, the authors present a genome-wide comparative analysis of Lactiplantibacillus pentosus and L. plantarum isolates autochthonous to cucumber fermentation. The objective of this study is to determine species specificity and to provide insights into niche adaptation.

This is an interesting study providing relevant knowledge on Lactiplantibacillus pentosus isolates autochthonous to cucumber fermentation.  

Specific comments

Please check carefully the references, many references are present in the text but not in the list and vice versa

-          Following references are in the text but not listed at the end of the manuscript:

Calero-Delgado et al., 2021

Ucar et al., 2020

Zanoni et al., 1987

Bringel et al., 1996

Barrangou, 2019

-          Following references are listed at the end but not present in the text of the manuscript:

Calero-Delgado et al., 2019

Deo et al., 2019

Kanehisa and Goto, 2000

Kanehisa, 2019

Kanehisa et al., 2021

Nurk et al., 2013

Round 2

Reviewer 1 Report

Dear Editors, 

The authors have made all requirements and corrections in the manuscript. The manuscript is now ready for publication

The language of the manuscript is good and it is easy for the reader.